# Costs and Consequences of Traffic Fines and Fees: A Case Study of Open Warrants in Las Vegas, Nevada

**Foster Kamanga** , **Virginia Smercina** , **Barbara G. Brents \*** , **Daniel Okamura** and **Vincent Fuentes**

Department of Sociology, University of Nevada, Las Vegas, NV 89154-5033, USA;
kamanga@unlv.nevada.edu (F.K.); smercina@unlv.nevada.edu (V.S.); okamud1@unlv.nevada.edu (D.O.);
fuentv3@unlv.nevada.edu (V.F.)
\* Correspondence: barb.brents@unlv.edu

**Abstract:** Traffic stops and tickets often have far-reaching consequences for poor and marginalized communities, yet resulting fines and fees increasingly fund local court systems. This paper critically explores who bears the brunt of traffic fines and fees in Nevada, historically one of the fastest growing and increasingly diverse states in the nation, and one of thirteen US states to prosecute minor traffic violations as criminal misdemeanors rather than civil infractions. Drawing on legislative histories, we find that state legislators in Nevada increased fines and fees to raise revenues. Using descriptive statistics to analyze the 2012–2020 open arrest warrants extracted from the Las Vegas Municipal Court, we find that 58.6% of all open warrants are from failure to pay tickets owing to administrative-related offenses—vehicle registration and maintenance, no license or plates, or no insurance. Those issued warrants for failure to pay are disproportionately for people who are Black and from the poorest areas in the region. Ultimately, the Nevada system of monetary traffic sanctions criminalizes poverty and reinforces racial disparities.

**Keywords:** monetary sanctions; fines and fees; court fees; criminal justice policy; racialized criminal justice policy; racialized traffic fines and fees; open warrants; traffic violations; criminalizing poverty; traffic tickets; mass incarceration

## 1. Introduction

The number of fines and fees imposed by local justice systems has increased dramatically since the 1980s in the US (Foster 2017), amounting to what Su (2020) labels as "taxation by citation". While some find that increasing monetary sanctions discourage traffic violations, there is increasing evidence that local governments across the globe rely on traffic related monetary sanctions to fund services (Singla et al. 2020; Su 2020; Garrett and Wagner 2009; Makowsky and Stratmann 2009).

Nevada is one of thirteen US states to prosecute minor traffic violations as criminal misdemeanors rather than civil infractions. Failure to pay a fine or a fee in Nevada results in what is called a bench warrant, wherein the judicial system asks for a person to be brought before a judge. While police do not actively hunt down these individuals, they can and frequently do arrest individuals in any situation where they encounter them. Given evidence that monetary sanctions foster inequality and disproportionately impact the poor and communities who may already suffer racial or ethnic discrimination, we ask: does Nevada's system of adjudicating minor traffic offenses amount to taxing poor and racially marginalized communities? Given recent policy concerns regarding the growth of and inequality in carceral systems, it is important to understand how traffic sanctions contribute to mass incarceration.

Nevada is an important place to examine the context and impact of increasing fines and fees. Since the 1990s, Nevada, and Las Vegas particularly, have been among the fastest growing areas in the USA. Job creation and economic growth in its leisure and tourism industries outpaced the nation, while intense urbanization and immigration have

transformed Nevada's demographics. Its white majority population in the past 30 years has become a white minority. Nevada is also a state where anti-tax sentiments run high. It is one of nine states with no state income tax, and it prides its economic growth on low taxes. Thus, dramatic population growth and a demographic shift have created dramatically increased demands on court systems, at the same time decreasing federal funding for courts, and anti-tax politics have hampered funding streams. In this context, it is important to examine legislative rationales for increasing traffic fines and fees, and inequality in who is most impacted by these increases.

While research is beginning to examine racial disparities in traffic stops, and in monetary sanctions generally, there is little research on demographic differences, especially economic inequality, in the warrants generated by failure to pay. In this study, we look at (RQ1) who suffers the impact of increasing traffic fines and fees in Las Vegas, Nevada, by examining the distribution of outstanding bench warrants across different demographic groups. Given the importance of understanding context, we also explore (RQ2) why Nevada increased fines and fees. We find that Nevada's system of making it harder to pay traffic tickets and then criminalizing those who cannot has disproportionately punished poor and Black communities.

## 2. Why Assess Fines for Traffic Violations?

### 2.1. Fines and Fees System Makes Roads Safer

There is some research supporting the view that increasing the cost of traffic violations improves public safety. Increasing traffic fines and fees can discourage road users from speeding, running red lights, and driving under the influence of alcohol, etc. (Makowsky and Stratmann 2011; Luca 2015; Tay 2010).

Makowsky and Stratmann (2011) used data from traffic stops and citations in Massachusetts to understand the relationship between numbers of traffic tickets and motor vehicle accidents and accident-related injuries. They found that an increase in traffic fines and fees reduced road accidents. DeAngelo and Hansen (2014) showed that deaths and injuries increased by 12–29 percent after highway troopers were massively laid off in Oregon. The authors found that the presence of traffic police increases the probability that a bad road user would get a citation, implying that the presence of law enforcement indirectly reduces risks on the road. Collectively, evidence supporting the road-safety argument shows that a higher rate of traffic citations reduces the likelihood of repeated traffic violations, thereby making the roads safer.

### 2.2. Traffic Fines and Fees System as a State's Revenue Generating Instrument

Alternatively, there are many studies that show little effect of traffic fines and fees on safe driving and argue the system acts more as a revenue generating instrument. They argue that revenue collection may motivate the issuing of traffic tickets (Li et al. 2011; Makowsky and Stratmann 2009; Montare 2019; Martin 2018; Singla et al. 2020), with selective enforcement determining who gets charged with traffic fines and under what circumstances. Financial hardships, such as the 2007 Great Recession, and reductions in tax revenue in municipalities and states have motivated higher rates of fines and fees through traffic citations (Garrett and Wagner 2009; Martin 2018; Singla et al. 2020).

Two examples are studies in North Carolina (Garrett and Wagner 2009) and California (Su 2020), which found that counties hiked rates of traffic fines immediately after collecting less tax revenue in the previous year. Yet these same counties never lowered traffic fines when they saw increased tax revenues (Su 2020). Makowsky and Stratmann (2009) found that in Massachusetts the likelihood of receiving a speeding fine was higher in towns experiencing a fiscal crunch caused by a rejected increase in the property tax limit. They also found that the chance of receiving a fine from a local officer is 11 percent higher for a driver who resides outside of the municipality than a driver who resides in the municipality where they are stopped. In a study comparing policies and practices on monetary sanctions between Nevada and Iowa, Martin (2018) found that monetary sanctions on misdemeanors

were motivated by the need for revenue to fund state services more than punishment that could lead to safer roads.

Put together, selective targets (nonresident drivers), increasing traffic citations due to declining funding, and reduced tax collections show that traffic citations are about generating revenue more than efforts to promote road safety. In the last decade, researchers and policy makers have sought to examine, expose, and eliminate the unequal effects of fines and fees, both in the US and globally. Scholars, legislators, and advocates have looked at how and why monetary sanctions are imposed, the specific ways courts impose fines on individuals, and how this impacts particular communities.

### 2.3. Fines and Fees Compound Economic and Racial Inequalities

Transparent and fair processes are especially important when traffic fines and fees are disproportionately levied against certain groups within the general population. The most common form of fine and fee revenue is the monetary fees that come from traffic violations. Research highlights that racial minorities are disproportionately affected by law enforcement and are overrepresented in traffic stops, citations, and frisks (United States Department of Justice 2015; Farrell et al. 2004; Norris et al. 1992; Pierson et al. 2020). More than a quarter of the 135 unarmed Black men and women killed by police since 2015 were killed during traffic stops, an NPR investigation has found (Thompson 2021). Fines are disproportionately imposed on poor individuals from communities of color (Alexander 2011; Burton and Lynn 2019).

Recently, Pierson et al. (2020), examined a dataset of over 60 million patrols conducted throughout 20 states between 2011 and 2015. The authors look at racial disparities in stop rates and post-stop outcomes. They find that Black drivers are stopped at an increased rate, compared to white drivers, relative to their share of the driving-age population. However, Hispanic drivers are less likely than whites to be stopped for traffic violations. Both Black and Latino drivers are more likely to be arrested, searched, or ticketed if they are stopped, compared to white drivers.

Earlier, Norris et al. (1992) looked at 213 police stops and 319 persons, and found that persons racialized as Black by police were more than two and a half times as likely to be stopped by police than their proportion in the population would suggest. Further, these stops disproportionately affected young Black men under thirty-five, and such stops disproportionately were carried out under general suspicion rather than enforcing an obvious violation of the law. Once stopped, the differences in people's comportment and in police treatment of those whom they stopped were negligible. However, because of their disproportionately high rate of stoppage, Black people were still more highly surveilled and issued formal sanctions than their incidence in the general population would expect.

One additional way the racialized fines and fees system plays out is through pretextual stops, which concerns how race and driver characteristics affect general suspicion for criminal activity (Miller 2010; Farrell et al. 2004). Using self-report data from telephone surveys of drivers in North Carolina, Miller (2010) draws from a reference frame of DMV records of 2620 Black and white residents who renewed a driver's license. He finds that local police are more likely to use racial status and youth as measures to justify increased scrutiny.

In a 2017 study by Sances and You, the authors found that cities relied more on fines and fees for revenue if they had a larger Black population. Additionally, cities with the highest Black populations assessed fees to citizens at more than twice the national average (Sances and You 2017). After police shot Michael Brown, a 2015 US Department of Justice report indicated that Ferguson, Missouri had increased its revenues through monetary sanctions each year while initiating revenue targets for officers to achieve. The police killings of Daunte Wright and Philando Castile in Minneapolis both occurred after traffic stops. Samuel DuBose, killed by a University of Cincinnati police officer while off-campus and unarmed, was stopped for missing a front license plate (Hunt 2017). Critics have raised

concerns of "cash-strapped systems targeting their own citizens using the policing powers of the government", (Mughan 2021).

Monetary sanctions have a huge impact on poor communities, exacerbating economic inequality. Drivers may have their licenses suspended, revoked, or suffer incarceration (Harris et al. 2016; Crozier and Garrett 2020), a practice that prevents people from accessing jobs. Currently, 44 states in the US suspend, revoke, or refuse driver license renewal if people have unpaid fines and fees. There are more than 11 million driver's license suspensions worldwide. In addition, drivers risk arrest, loss of voting rights (Fredericksen and Lassiter 2016; Fines and Fees Justice Center 2018), job loss, reduced housing and credit opportunities (Martin 2018; Beckett et al. 2008), and repeated incarceration and continued involvement with the criminal justice system (Martin 2018).

In addition, traffic-related fines and fees affect the relationship between the state, defendants of citation, and the whole citizenry. Tickets that are too costly lead to non-compliance, which is not only detrimental to budget forecasting (Hummel 2015) but also, more importantly, excessive traffic tickets and fines that are unfairly distributed weaken the trust between citizens and those who enforce the law (Nyberg et al. 2021; Su 2020; Bornstein and Tomkins 2015).

## 3. Methods

In this study, we examine the context and impact of increasing traffic fines and fees in Las Vegas, Nevada. We first look at the legislative history in Nevada of increasing monetary sanctions. Because unpaid traffic fines and fees turn into bench warrants, we then considered who is most impacted by traffic fines and fees when they cannot pay by examining the demographics of individuals and the violations incurred among those with outstanding traffic warrants in one of its courts, the Las Vegas Municipal Court.

*3.1. Data Sources*

3.1.1. 2017–2018 Nevada Legislative Interim Study Committee

To answer RQ2, why Nevada has increased fines and fees, we examined data gathered from a 2017–2018 Nevada Legislature Interim Committee Study and meeting notes by the Committee to Study the Advisability and Feasibility of Treating Certain Traffic and Related Violations as Civil Infractions. The Nevada State Legislature meets every other year for a 120-day session and adopts budgets and laws intended for the following two years. Between sessions, interim committees are created by members of the legislature to work on specific issues. These committees often receive public comment and publish reports, and they produce recommendations for the following legislative session. We examined reports and testimony from one such study specifically on the assessment of traffic fines and fees (Nevada Legislative Interim Study 2018). We situated this data in a historical context to look for patterns in and motives for legislation.

3.1.2. Las Vegas Municipal Court Open Warrant Data: 2012–2020

To examine the RQ1, who suffers the impact of increasing traffic fines and fees, we analyzed all outstanding bench warrants for misdemeanors that the Las Vegas Municipal Court issued from 2012 to 2021 and filtered the data to include traffic-related violations only. Courts issue "bench warrants" against road users who have not paid and/or failed to appear in court for traffic citations. Bench warrants are orders issued by a judge instructing police to arrest people for defying court requirements or rules. A traffic citation becomes a bench warrant after the defendant fails to pay the fine and fees and after failing to appear before the court in Nevada. The warrant is sent by mail to the address that the defendant reported. In Las Vegas, the court issues an additional USD 200 warrant fee. Bench warrants do not expire until the fine is paid.

When a judge issues a warrant, it is a matter of public record, and many cities have searchable websites or lists (as PDF documents, for example) that allow people to check if there is an active warrant out for their arrest. Defendants' information may not always be

kept together in a convenient format such as a geotagged database. Instead, they are often in a form that serves the government and public need to look up active warrants.

We used quantitative data on outstanding warrants that we extracted from the "City of Las Vegas Marshal—Warrant Search" website on 3 January 2021 (https://secure3 .lasvegasnevada.gov/ewarrantlookup/, accessed on 3 January 2021). These warrants were issued by the Las Vegas Municipal Court between 2011 and 2021 to individuals who were ticketed or arrested within the city limits of Las Vegas, Nevada. These were the only warrant years that were publicly available. While there are other courts in Nevada, Las Vegas is the largest city in the state, and the Las Vegas Metropolitan area contains three-fourths of the state's population. The Las Vegas Municipal court data was the only data publicly available in the Las Vegas area with the necessary information for our assessment, and we will discuss this below.

Our analysis focused on warrants that the court issued for the period 2012 to 2020. We excluded warrant data for the years 2011 and 2021 because we observed that there were too few warrants issued during those time periods at the time of data extraction in January 2021. The warrant data we extracted contained 403 charges (with some listed redundantly), which we collapsed into 24 charge categories (see Table A1), N = 102,466. Those 24 categories were then compressed by charge type into eight categories for analysis. (See Table A2). We considered the first 7 categories as traffic-related and the 8th charge category as a non-traffic charge. As the appendix section shows at the end of this report, the list of 8 categories of charges include insurance, driver's license, vehicle registration, vehicle condition, moving violations, parking, DUI, and non-traffic. Besides traffic charges, the warrant data contains defendant demographics, including defendant's known address, ZIP code (postal code), race, sex, age, and bail amount.

*3.2. Data Analysis*

To estimate income, which was not in the warrant data, we used the Google Maps API (https://developers.google.com/maps/documentation/geocoding/overview, accessed on 16 December 2020) to geocode defendants addresses into corresponding latitudes and longitudes, which we used to generate maps in Tableau Public. We linked the geographic coordinates to their corresponding census blocks, which we mapped from The Federal Communications Commission (FCC) API at https://geo.fcc.gov/api/census/ (accessed on 16 December 2020). We used census blocks to link the warrant dataset to data that we downloaded from the US Census's American Community Survey, 2019, which estimates the median income of people in block groups. Estimates from the US Census American Community Survey allowed us to approximate indirectly the income levels attained by road users that received bench warrants.

To analyze the demographic and infraction data, we used SPSS 27 to estimate univariate statistics. We discuss these in the findings section to understand who is impacted by the traffic fines and fees when they cannot pay and appear before the court in Las Vegas, Nevada.

**4. Results**

*4.1. Why Has Nevada Increased Fines and Fees? A Legislative History*

The data we examined, as well as a study by Martin (2018), indicates that Nevada raised traffic fines and fees to meet budget shortfalls, not to discourage traffic violations. A 2017–2018 interim study by the Nevada State Legislature cites the 1980s recession as its primary reason for beginning to raise fines and fees for traffic violations. That recession led Congress to cut approximately USD 40 billion from the 1982 budget, which limited the amount of funds that Nevada and other states could use for their justice systems (Nevada Legislative Interim Study 2018). Nevada's courts and criminal legal systems had been funded primarily by the federal Law Enforcement Assistance Administration (LEAA) (Martin 2018).

In 1983, the Nevada Legislature replaced these lost federal funds by authorizing a USD 10 Administrative Assessment (AA) on all misdemeanors. The fee was distributed in the following ways:

1.  USD 1 for city/county juvenile court(s);
2.  USD 3 for Municipal/Justice court(s);
3.  USD 5 to Supreme Court/The Administrative Office of the Courts (AOC)
4.  USD 2 for AOC;
5.  USD 2 for Uniform System of Judicial Records (USJR);
6.  USD 1 Judicial Education;
7.  USD 1 for Peace Officer Standards and Training (POST).

Legislative sessions after 1983 continued to increase AA fees. During the 1985 session, Nevada legislature reallocated the USD 1 additional fees from local courts to the Supreme Court, thus funding the Supreme Court's activities from AA funds (Nevada Legislative Interim Study 2018).

The AA fee also helped fund additional criminal justice functions (Martin 2018). In 1987, the Nevada Legislature raised AA fees from USD 10 to USD 100 to fund the following expansion projects and upgrade technology: the legislature allocated almost USD 89 of the USD 100 to the executive branch, including the Criminal History repository, the Nevada Division of Investigation (NDI), the computerized Nevada Highway Patrol (NHP) switching system, and the Victims of Crime Fund (Nevada Legislative Interim Study 2018).

From the 1980s to 2010, Nevada experienced some of the fastest population growth in the country and was rapidly expanding and upgrading its services. The additional AA funds helped develop the Highway Patrol Mobile Data Computer Project. Technological expansion led to the need for training personnel and other services (Nevada Legislative Interim Study 2018). The state simply became reliant on traffic fees, and Table 1 shows that subsequent legislative sessions continued raising AA fees.

**Table 1.** Administrative Assessment Fund Increases.

| Year | Outcome of the Legislative Session |
|------|-------------------------------------|
| 1991 | Redistributed AA funds<br>51% to supreme court<br>49% the executive branch |
| 1995 | Authorized county/city to pass an AA fee of USD 10 for facilities |
| 1997 | Increased AA schedule from USD 100 to USD 105<br>Authorized USD 2 and USD 7 for Juvenile and Justice/Muni courts respectively |
| 2001 | Authorized Supreme court to receive higher than 51% of AA revenue<br>Added 'Advisory Council for Prosecuting Attorneys (AG's Office)' to the recipients of AA funds within the executive branch |
| 2003 | Increased AA from USD 105 to USD 115<br>Authorized USD 7 as specialty court AA |
| 2007 | Reduced AA allocation to supreme court to 48%<br>Authorized the 'other 12% to fund specialty courts' |
| 2010 | Increased AA funds from USD 115 to USD 120<br>Authorized that the funds collected from the USD 5 increment be sent directly to the state general fund |

As we can see, while federal budget cuts started the trend toward increasing fines and fees, legislators funded judicial services needed to meet Nevada's growth not by increasing taxes, but through maintaining and increasing monetary sanctions. As we show below, this taxation by citation has disproportionately impacted those receiving traffic citations for administrative infractions, most often affecting poor and Black populations.

*4.2. What Violations Receive the Most Warrants?*

Figure 1 is a map of Nevada showing circles that represent the ZIP codes of the home addresses reported by individuals who have received bench warrants from Las Vegas Municipal Court between 2012 and 2020. The larger the circle the greater the number of open warrants within that zip code. From the map, the largest circle is in ZIP code 89108, indicating the most open warrants. Las Vegas Municipal Court covers traffic tickets issued within the Las Vegas city limits. Note: only outstanding warrants containing ZIP code-level data were geocoded. As a result, of 44,373 open warrants 34,338 (77.3%) were geocoded.

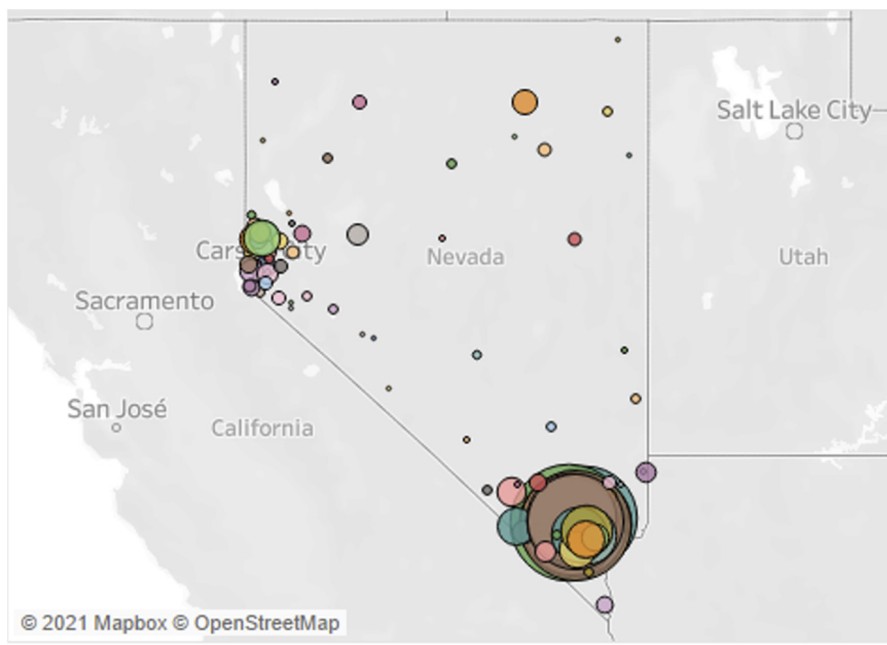

**Figure 1.** Map of reported ZIP codes with open bench warrants from Las Vegas Municipal Court between 2012 and 2020. Circle size increases with the number of warrants in each ZIP code.

Figure 2 shows that, from 2012 to 2020, relatively few bench warrants were due to non-traffic charges as compared to traffic charges. Non-traffic charges include misdemeanors for theft, battery, domestic violence, loitering, etc. Most bench warrants (83.3%) were due to traffic charges, the majority (58.6%) of which were for administrative violations compared to 24.7% directly connected to behavioral violations in Nevada. Table 2 shows categories within each major grouping, with driver's license suspensions, vehicle registration, and lack of insurance being the most common administrative violations, all violations typically resulting from low incomes.

**Table 2.** Behavioral vs. Administrative Traffic Charges: 8 Categories.

| Major Categories | # of Warrants per Violation | Warrants % |
|---|---|---|
| Administrative Traffic Charges | 59,115 | 58.6 |
| Insurance | 15,457 | 15.3 |
| Drivers' License | 26,234 | 26 |
| Vehicle Conditions | 1957 | 1.9 |
| Vehicle Registrations | 15,467 | 15.4 |
| Behavioral Traffic Charges | 24,828 | 24.7 |
| Moving Violations | 16,085 | 16 |
| Parking | 6515 | 6.5 |
| DUI | 2228 | 2.2 |
| Total Traffic | 83,943 | 83.3 |
| Non-Traffic | 16,801 | 16.7 |
| Total: | 100,744 | 100% |

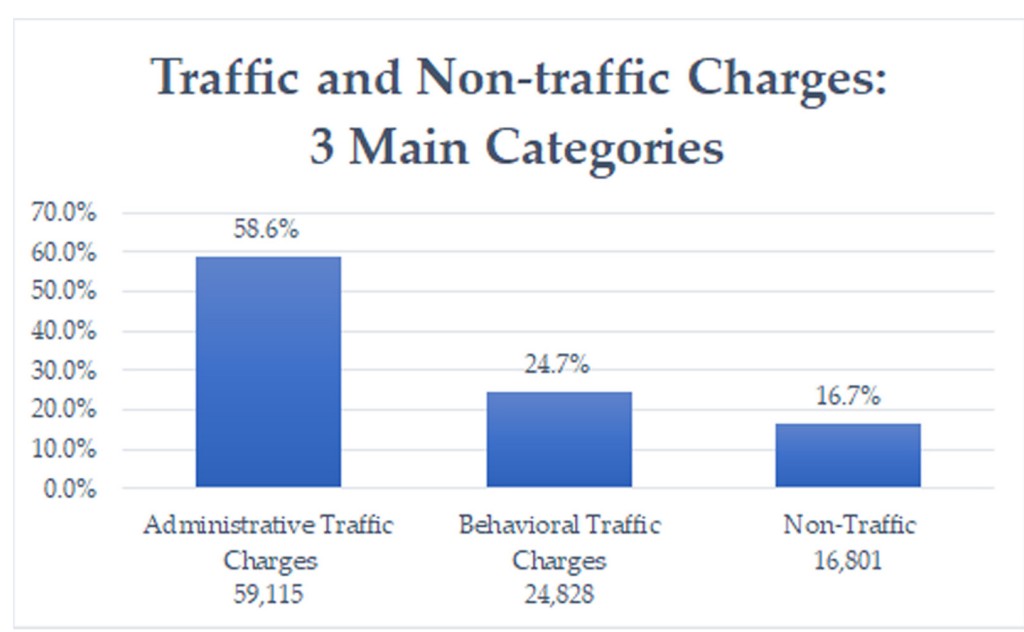

**Figure 2.** Traffic and Non-traffic Charges: 3 Main Categories.

*4.3. Who Is Impacted? Economic Inequality*

Figure 3 is a map of the Las Vegas Metropolitan area and shows the median household income in the ZIP code for listed addresses of individuals with open bench warrants from Las Vegas Municipal Court between 2012 and 2020. In red are ZIP codes with the most open warrants. In blue are the wealthiest ZIP codes in Clark County. Table 3 breaks down each ZIP code by median income and percent of open warrants.

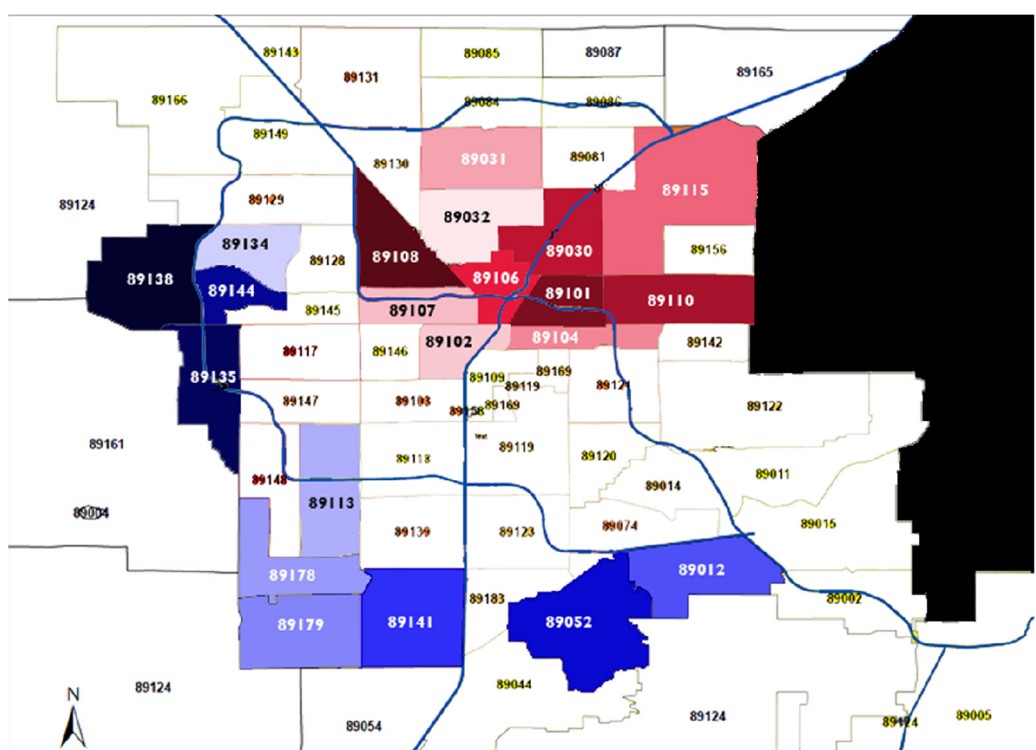

**Figure 3.** Median Income and Number of Warrants by ZIP Code Clark County.

**Table 3.** Figure 3 Key—Open Warrants by ZIP Codes in Clark County, Nevada (City of Las Vegas Median Household Income USD 56,354).

| ZIP Codes with Most Open Warrants | | | Wealthiest ZIP Codes | | |
|---|---|---|---|---|---|
| **Zip** | **Median Income** [1] | **Open Warrants %** | **Zip** | **Median Income** | **Open Warrants %** |
| 89108 | USD 46,165 | 8.9 | 89138 | USD 120,759 | 0.1 |
| 89101 | USD 25,310 | 7.2 | 89135 | USD 94,821 | 0.3 |
| 89110 | USD 44,415 | 5.6 | 89144 | USD 88,750 | 0.3 |
| 89030 | USD 36,275 | 5.5 | 89052 | USD 85,021 | 0.2 |
| 89106 | USD 29,906 | 5.2 | 89141 | USD 89,649 | 0.4 |
| 89115 | USD 39,412 | 4.8 | 89012 | USD 81,992 | 0.2 |
| 89104 | USD 36,448 | 4.3 | 89179 | USD 99,662 | 0 |
| 89031 | USD 66,270 | 4.1 | 89178 | USD 88,517 | 0.4 |
| 89107 | USD 44,234 | 4.1 | 89113 | USD 72,479 | 0.4 |
| 89102 | USD 36,729 | 3.9 | 89134 | USD 69,461 | 0.3 |
| 89032 | USD 60,294 | 3.5 | Total Warrants | | 2.60% |
| Total Warrants | | 57.10% | | | |

[1] Data are from 2019 inflation adjusted dollars from the American Community Survey 2019 5-year estimates. Nevada Income Statistics https://www.incomebyzipcode.com/nevada, accessed on 12 March 2021.

The wealthiest ZIP codes make up 2.6% of all open warrants. Las Vegas Municipal Court deals with warrants for traffic violations in the city limits of Las Vegas, and Las Vegas has a median income of USD 56,354, only slightly less than the median income of Clark County as a whole (USD 59,340). Nonetheless, of the ZIP codes with the most open warrants (57.1%), all but two have median incomes below USD 56,354. Several are among the poorest ZIP codes in Clark County.

For a finer-grained analysis of the likely income of individuals with bench warrants, we examine the median income of census block groups containing the addresses of individuals with bench warrants. Census block groups are typically 3–5 times smaller and have fewer people (250–550 housing units) than ZIP codes.

Table 4 shows that a majority of the defendants' (58.5%) addresses were in block groups whose estimated household median income was USD 49,000 a year or below. The median household income in the City of Las Vegas, the jurisdiction where traffic infractions occurred, is USD 56,354.

**Table 4.** Distribution of Income of Based on Defendant's Address [1].

| Income Range | # of Open Warrants | Warrants % |
|---|---|---|
| USD 49,999 and below | 41,412 | 58.50% |
| USD 50,000 to USD 99,999 | 27,227 | 38.50% |
| USD 100,000 and above | 2115 | 3.00% |
| Total: | 70,754 [2] | 100% |

[1] Income is based on the household median income reported by the US Census American Community Survey 2019 report of the Clark County block groups containing the defendant's address, not an individual's actual reported income. [2] Total excludes 31,712 missing cases. These are open warrants with no ZIP code associated. In some of these cases, individuals may have no known home address. Missing cases are omitted from the analysis. Total open warrants were 102,466.

Most warrants are issued to individuals in some of the poorest areas of the Las Vegas Valley. As Figure 4 shows, the majority of defendants were from Census block groups that had household median incomes below the median for the city of Las Vegas. By contrast, an estimated 3% of defendants lived in block groups with a household median income of

USD 100,000 and above. The data demonstrate that the majority of warrants are issued to those least able to pay.

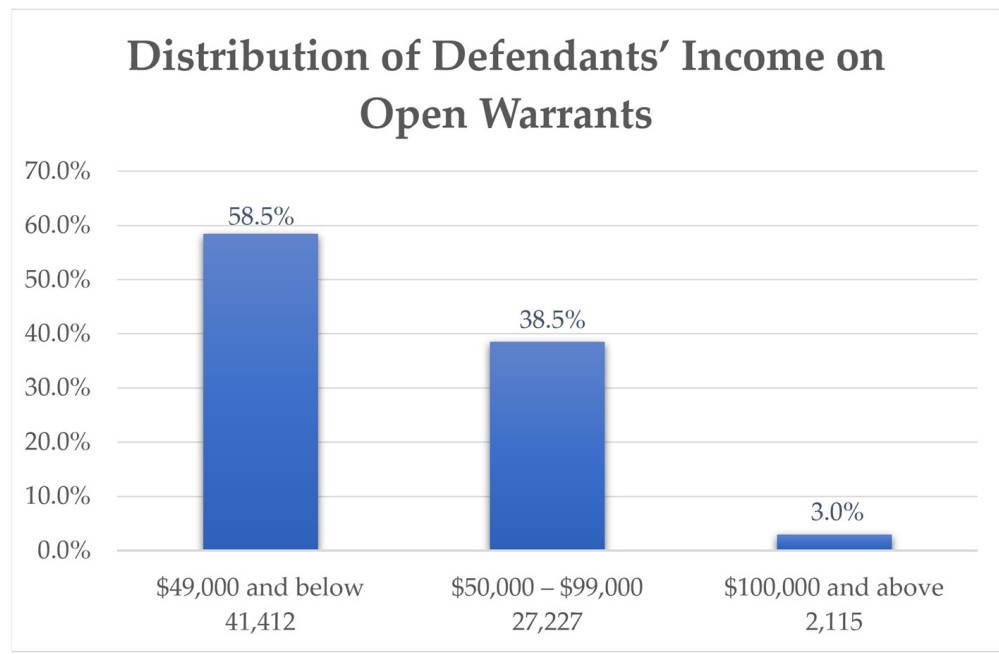

**Figure 4.** Distribution of Defendants' Income on Open Warrants.

### 4.4. Who Is Impacted? Racial Inequality

According to the US Census, Blacks make up 13.1% of the Clark County population, while whites make up 69.5%. Yet Table 5 and Figure 5 below show that, from 2012 to 2020, Blacks are very disproportionately represented among those with open warrants. Black individuals comprise 44.7% of those who have open warrants as compared white (31.1%). Hispanics have proportionally fewer outstanding warrants (21.9%) compared to their numbers in the population.

Table 6 shows that people who are Black, Hispanic, or Asian/Pacific Islander are noticeably more likely to have warrants for administrative charges as compared to whites. Indian/Alaskan Natives are the least likely to have open warrants due to administrative traffic charges but much more likely to have warrants for non-traffic offenses.

**Table 5.** Racial/Ethnic Distribution on Open Warrants.

| Race/Ethnicity | # of Open Warrants | Warrants % | Clark County Population |
|---|---|---|---|
| Black | 43,627 | 44.70% | 13.10% |
| White | 30,341 | 31.10% | 41.70% |
| Hispanic | 21,321 | 21.90% | 31.60% |
| Asian/Pacific Islander | 2136 | 2.20% | 11.30% |
| Indian/Alaskan Native | 123 | 0.10% | 1.20% |
| Total: | 97,548 [1] | 100.00% | 98.90% |

Total excludes 4918 missing cases. These are open warrants with no race/ethnicity associated.

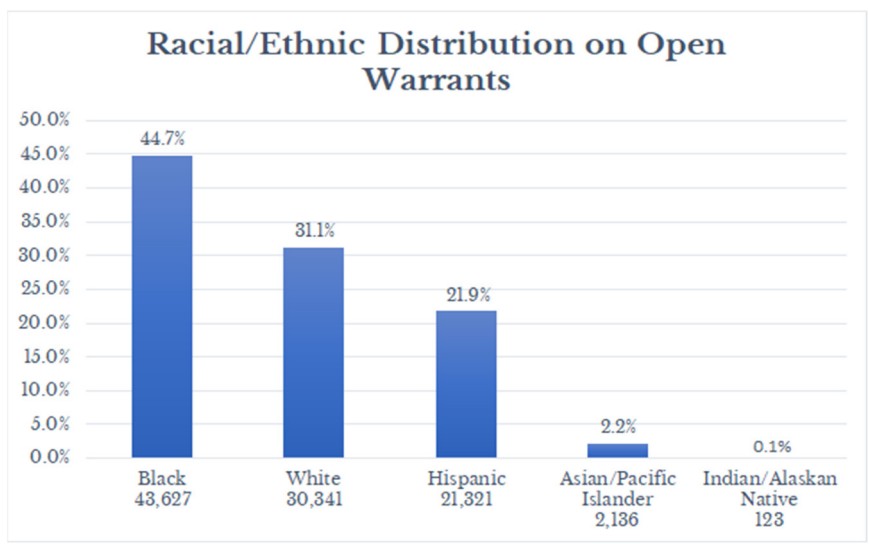

**Figure 5.** Racial/Ethnic Distribution on Open Warrants.

**Table 6.** Distribution of Open Warrants by Race/Ethnicity and Charge Type (N = 95,842) [1].

| | Black | White | Hispanic | Indian/ Alaskan Native | Asian/Pacific Islander |
|---|---|---|---|---|---|
| **Administrative** | 61.9% (26,606) | 54.4% (16,058) | 61.0% (12,853) | 34.2% (8) | 61.5% (326) |
| **Behavioral** | 23.2% (9957) | 24.1% (7106) | 27.5% (5789) | 25.7% (30) | 23.4% (488) |
| **Non traffic** | 15.1% (6492) | 21.6% (6367) | 11.4% (2408) | 40.2% (47) | 15.2% (318) |
| **Totals** | 100% (43,055) | 100% (29,531) | 100 (21,050) | 100% (117) | 100% (2089) |

[1] Missing values for either race/ethnicity or charge type were omitted, resulting in 6624 missing values.

*4.5. Who Is Impacted? Gender and Age*

The Table 7 and Figure 6 below show that there are more male (65.7%) than female (34.3%) defendants with outstanding warrants. Males are disproportionately represented, yet the US Census shows that there are slightly more females (50.1%) than males in Las Vegas.

The vast majority of individuals with open warrants from 2012 to 2020 in Nevada were in the 18–55 age group (89.6%). The largest category included those between 18–34 (47.1%). As Table 8 and Figure 7 show, individuals younger than age 17 were the smallest category of defendants.

**Table 7.** Open Warrants by Gender.

| Gender | # of Open Warrants | Warrants % |
|---|---|---|
| Female | 35,012 | 34.3 |
| Male | 67,026 | 65.7 |
| Total: | 1,020,381 | 100% |

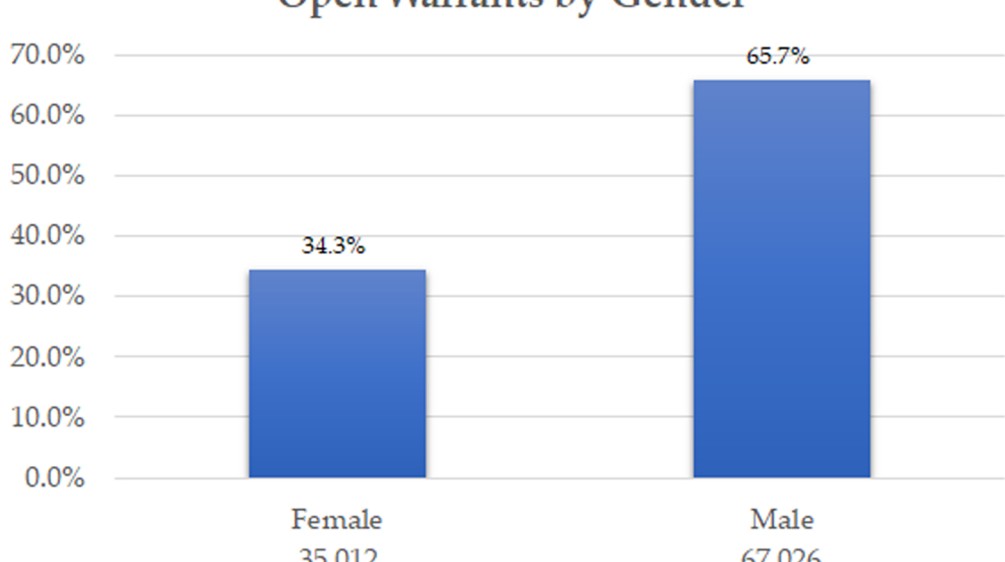

**Figure 6.** Open Warrants shown by Gender.

**Table 8.** Distribution of Defendants' Age on Open Warrants.

| Age Range | # of Open Warrants | Warrants % |
|---|---|---|
| 17 or younger | 137 | 0.10% |
| 18 to 34 | 48,280 | 47.10% |
| 35 to 55 | 43,561 | 42.50% |
| 56 to 75 | 9992 | 9.80% |
| 76 or older | 451 | 0.40% |
| Total: | 102,421 | 100% |

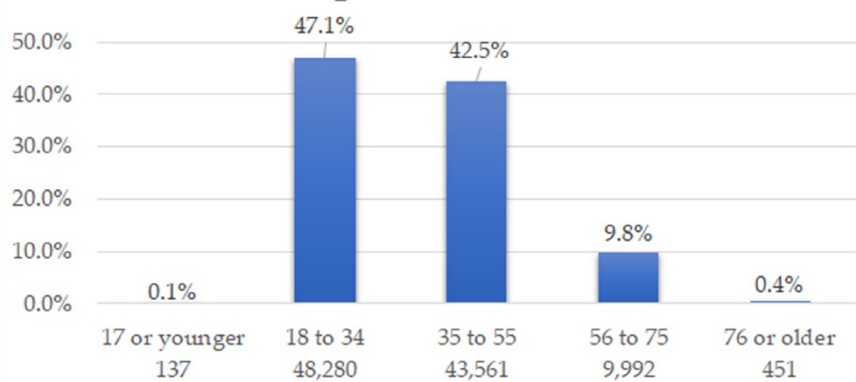

**Figure 7.** Distribution of Defendants' Age on Open Warrants.

## 5. Discussion

Our data reveal that the persons most commonly impacted by bench warrants were between the ages of 18 and 34 (47.1%), were more likely to be male (65.7%), from the poorest ZIP codes (58% from median income of USD 49,000 and below), and much more likely to be Black than any other ethnicity (44.7%). Poorer areas were identified as areas with incomes by ZIP code below the Las Vegas median income, some significantly below.

In the context of Nevada's legislative history of monetary sanctions and data from the Las Vegas Municipal Court, our results have troubling implications.

Financial shortfalls, not enhancing road safety, was the motivation for Nevada to increase fines, including those related to traffic offenses. The Nevada Legislature chose to adopt policies to increase Administrative Assessment (AA) fees to make up for lost revenue. They continued to raise fees and not raise taxes to fund state and municipal services as population grew.

Furthermore, the evidence suggests that AA fees fund state services unrelated to the courts dealing with most violations (Nevada Legislative Interim Study 2018). For example, during its 2013 session, the Nevada Legislature imposed a USD 3 AA fee on all offenses to expand the executive branch and to fund DNA testing of felony arrestees (Nevada Legislative Interim Study 2018). In the same legislative session, a USD 100 fee was authorized on misdemeanor DUI offenses in addition to the USD 120 AA fee. The USD 100 fee schedule was to "sunset in June of 2015", but the legislature continued authorizing the fee until 2017. The 2015 session allocated USD 558,000 to Nevada's supreme court, which had a shortfall in AA fee revenue. Martin (2018) found that, as the administrative assessments increased, the proportion that courts received from the state's general fund decreased. Our review of the data provided by the 2017–2018 Nevada Legislative Interim Committee Study provides no significant evidence that legislators intended the incremental increases in administrative assessment schedules to deter increased cases of bad driving. Thus, the Nevada criminal legal system depends financially on penalizing its citizens.

Further, our evidence shows that it is not moving violations that are the source of most warrants issued to those unable to pay these increasing fees. The most common traffic violations that led to a bench warrant were based on administrative violations such as the inability to pay for insurance, driver's licenses, or vehicle registration. In other words, if one is too poor to pay for vehicle registrations, they incur additional fines and fees and risk jail. Nevada's system of taxation by citation increases economic inequality among low-income individuals (Sances and You 2017).

At a time when Nevada was becoming demographically diverse, choosing to tax by citation to fund services contributes to the structural racism that fuels mass incarceration. The most troubling finding from our study is how racially inequitable this system of traffic fees is. The current system of "taxation by citation" in Nevada is extremely inequitable, disproportionately impacting Black citizens in particular. Previous research highlights how racially marginalized groups are overrepresented in traffic stops (United States Department of Justice 2015; Farrell et al. 2004; Norris et al. 1992; Pierson et al. 2020). It appears that the state's low taxes add to the burden felt by racially marginalized citizens and contributes to racialized mass incarceration and the disproportionate number of black men who are justice system impacted.

*Future Research*

The fiscal implications of fines and fees has real-world consequences to not only the individuals who incur them but also to American society overall. The impact of fines and fees in Nevada is a story that is still evolving. More quantitative as well as qualitative research is needed. We need to know what proportion of traffic citations become warrants. We need to know who is getting cited and for how much. More information is needed to better understand who can pay traffic citations versus who cannot. Finally, we need a better understanding of exactly how much money is collected from traffic fines and fees, and what monetary sanctions are used for.

## 6. Limitations

Data on the impacts of monetary sanctions are notoriously difficult to gather due to haphazard and uneven records kept by law enforcement (Martin et al. 2018), and we were only able to use data from one court. Many courts in Nevada operate using antiquated software systems that limit reporting capabilities. For example, while the North Las Vegas

Municipal Court case management system does capture data of interest to this project, it is incapable of extracting data into a report format. Additionally, the charges may not be uniform across jurisdictions. For our research, data collection attempts were made over a ten-week period from September 2020 through January 2021, focusing on Clark County, Nevada. Due to the COVID-19 pandemic, the staff for the district courts were working restricted hours and were unable to assist with data requests.

Additionally, we were unable to obtain detailed information on who is issued tickets. Some US cities do maintain public databases, but doing so often comes down to staffing and resources. The more sophisticated a city's public-facing website, the more labor intensive and expensive it is to construct and maintain, which typically requires a larger population to form a larger tax base (Lidén 2017).

Reliable data allowing for sufficient transparency in the use of public funds is a critical need. Pierson et al. (2020) assert that states should collect individual level stop data that have the following measures: date and time of the stop, location, race, gender, and age of driver, the stop reason, whether a search was conducted, and a short narrative written by the officer. The authors cite New York City's UF-250 form for pedestrian stops as an example of how to utilize this level of data.

Law enforcement agencies must continue to make their data accessible to researchers and to the public. It is also recommended that police departments regularly analyze the data they collect and ambitiously design statistically informed guidelines informing their decisions. Providing this research to the public along with their coding process would help to bring much needed transparency to the issue of public relations with police.

## 7. Conclusions

In this study, we have demonstrated dramatic inequality in who is impacted by how the criminal-legal system responds to declining government funding. In Nevada, the 1980s recession was the catalyst for increasing fines and fees as it steered Congress to significantly limit funding for state judicial systems, including Nevada (Nevada Legislative Interim Study 2018). It is likely other states and indeed other nations are making similar decisions in a global context of declining government revenues.

As dramatic population growth increased demands on court systems in Nevada, the state's anti-tax politics hampered funding streams. The increased reliance on users of the criminal-legal system came at the same time as a dramatic demographic shift in the state. Between 1980 and 2020, Nevada become majority Black and Latinx at a time when criminal-legal institutions nationwide were increasingly surveilling and jailing non-white communities, amounting to what some have called, "the new Jim Crow" (Alexander 2011). Our findings show that Nevada's system of fines and fee makes it harder to afford traffic tickets and then criminalizes those who are unable to pay. Poor Black communities are disproportionately punished under this system, and this has fed into the trend toward racialized mass incarceration. Thus, the system not only increased economic inequality, it also exacerbated structural racism.

In summary, our data strongly indicates not only the economic and racial inequality in hardships Nevada's system of criminalizing failure to pay monetary sanctions but also highlights an overarching perversion of how local government is funded while exposing larger societal dangers this system can cultivate. We have told the story of how this has happened in one US state, but we know that increasing reliance on fines and fees can have similar negative implications in vulnerable communities across the globe.

The US Constitution forbids punishing people based on their economic status (Garrett et al. 2020). Sterling (2019) states that "the current system for adjudicating misdemeanors looks more like a criminal processing system meant to generate revenue than a criminal justice system meant to generate fairness," (p. 1). This directly contradicts the edict of the judicial/criminal justice system, which focuses on behavioral correction and criminal reformation. The goal of social justice reform should be to create, "an equitable system

that upholds human rights and the dignity of people regardless of background," (Varghese et al. 2019, p. 683).

The current fines and fees system in Nevada, as in other states, is a consequence of the federal government shifting the cost burden of local services to states who are otherwise unprepared to raise funds, unwilling to raise general taxes, or who may not understand how additional service fees disproportionately fall on the most marginalized communities.

*Toward Criminal Justice Reform*

There is good news. The information collected and analyzed in this study, with support of the Fines and Fees Justice Center, helped pass legislation in the 2021 Nevada Legislature to decriminalize traffic tickets. As part of broader efforts toward criminal justice reform, Assembly Bill 116 will downgrade minor traffic violations from misdemeanors to civil infractions that do not require jail time. The law will not go into force until 1 January 2023 to allow local jurisdictions to adjust their processing systems (Lyle 2021).

The law does not eliminate the fines and fees associated with traffic violations but ends the practice of issuing a warrant for arrest based on a person's inability to pay. This change also avoids criminalizing and further hindering those unable to pay these fees. While this will sever one connection between being unable to pay a fine and incarceration, improving a systemically unequal and racially biased practice, this does not end taxation by citation. Policy makers justified the bill as a net saving. Jail time is estimated to cost USD 400 per incarceration, and costs of collections are high. There are currently 270,000 open warrants, potentially USD 1.35 million, and it is unclear if these will be pursued. Nevada is still heavily dependent on monetary sanctions to fund its judicial system, and until taxation by citation ends, the system will continue to foster economic inequality and compound racial discrimination.

**Author Contributions:** Conceptualization, B.B, F.K., V.S. and D.O.; methodology, F.K, V.S. and D.O.; validation, V.S. and F.K.; formal analysis, V.S., F.K. and D.O.; investigation, F.K., V.S. and B.G.B.; writing—original draft preparation, V.S., V.F., F.K. and B.B; writing—review and editing, B.G.B., F.K., V.S. and D.O.; visualization, V.S. and B.G.B.; supervision, B.G.B.; project administration, B.G.B. All authors have read and agreed to the published version of the manuscript.

**Funding:** This research received no external funding.

**Institutional Review Board Statement:** The study was conducted according to the guidelines of the Office of Research Integrity—Human Subjects and was approved as exempt by the Social/Behavioral Institutional Review Board (IRB) of the University of Nevada, Las Vegas (#1715073-3, Fines, Fees, and Inequality; 25 February 2021).

**Informed Consent Statement:** Not applicable.

**Data Availability Statement:** Data can be made available upon request.

**Acknowledgments:** We thank Roger Pharr of the New York University's Public Safety Lab for invaluable assistance collecting and analyzing data. We thank the City of Las Vegas Municipal Court for recognizing the need to record reliable and searchable data. We thank Leisa Moseley and the Fines and Fees Justice Center for their amazing encouragement and assistance in this project and in revising the law in Nevada.

**Conflicts of Interest:** The authors declare no conflict of interest.

## Appendix A

**Table A1.** Data's charge categories reduced to 24 new charge category labels.

| New Value | New Label | Old Label (of Charge) |
|---|---|---|
| 1 | Insurance | NO INSURANCE/SECURITY +<br>NO PROOF OF INSURANCE + |
| 2 | Driver license | CHANGE NAME/ADD ON DRIV LIC—IN 30 DAY +<br>DRIVE ON CANCELLED DRIV LIC +<br>DRIVE ON REVOKED DRIV LIC +<br>DRIVE ON SUSP/CANC/REV DRIVE LIC +<br>DRIVE ON SUSPENDED DRIVERS LICENSE +<br>DRIVERS LICENSE—URRENDER ON DEMAND +<br>DRIVING WITHOUT VALID LICENSE +<br>DRIVING WITHOUT VALID LICENSE—EXPIRED +<br>EMPLOY UNLICENSED DRVER/DRIVE MTR VEH +<br>MOTORCYCLE DRIVERS LICENSE REQUIRED +<br>NO DRIVERS LICENSE +<br>NO DRIVERS LICENSE IN POSSESSION +<br>NO MTRCYCLE DRIV LIC IN POSS—ADULT/JUV +<br>NO NEVADA DRIVERS LIC WITHIN 30 DAYS +<br>POSSESS ALTERED DRIV LIC +<br>POSSESS REVOKED DRIV LIC +<br>POSSESS SUSPENDED DRIVERS LICENSE +<br>USE/POSS SUSP/CANC/REV DRIV LIC +<br>VIOLATE INSTRUCTION PERMIT REQUIREMNTS +<br>VIOLATION OF RESTRICTION ON LICENSE +<br>UNLAWFUL TRANSFER OF LICENSE PLATES +<br>MORE THAN ONE DRIVERS LICENSE<br>NO COMMERCIAL LICENSE W/VALID ENDORSE |
| 3 | Vehicle conditions general/Road worthiness | BRAKE MAINTENANCE +<br><br>BRAKES +<br>BRAKES—EVERY MOTOR VEHICLE MUST HAVE +<br>BRAKES STOP WITHIN CERTAIN DISTANCE +<br>EXHAUST SYS—EXTEND PAST REAR/SIDE VEH +<br>EXHAUST SYSTEM—MUST BE GAS TIGHT +<br>ALL MOTOR VEHICLES MUST HAVE MIRRORS +<br>ALL MOTOR VEHICLES MUST HAVE MUFFLERS +<br>FENDERS REQUIRED +<br>MUST HAVE HORNS AND WARNING DEVICES +<br>MUST HAVE WINDSHIELDS—NOT DEFECTIVE +<br>TIRES—TREAD DEPTH—UNSAFE +<br>UNSAFE VEH—NOT EQUIPPED AS REQUIRED +<br>WINDSHIELDS/WINDOWS NOT OBSTRUCTED +<br>MIRRORS—TWO REQ NOT LESS 3" IN DIAMETR<br>WINDSHIELD WIPERS—MUST HAVE AND MAINTAIN |
| 4a | Lamps | 2 TAIL LIGHTS REQ/LOCATION AND DISTANCE +<br>FLASHING RED OR YELLOW SIGNAL +<br>HEAD LAMPS HOURS OF OPERATION +<br>HEAD LAMPS—2 REQUIRED/LOCATION ON VEH +<br>HEAD LAMPS—AT LEAST ONE REQ—PROPER LOC +<br>HIGH AND LOW BEAM USE—FAIL TO DIM +<br>LAMPS ON PARKED VEHICLES +<br>MOTORCYCLE DEFECTIVE HEADLAMPS +<br>MTR CYCLE HEADLAMPS—TIMES/OPERATION +<br>REFLECTOR AND CLEARANCE LAMP LOCATIONS +<br>STOP LAMPS REQUIRED +<br>TAIL LAMPS—REQUIRED WHEN HEADLIGHTS ON + |

**Table A1.** *Cont.*

| New Value | New Label | Old Label (of Charge) |
|---|---|---|
| 4b | Light Color/Signals | COLOR OF LAMPS +<br>DISPLAY BLUE LIGHTS ON NON-EMERGENCY VEHICLE +<br>REFLECTORS—POSITION AND SIZE +<br>EXTRA LIGHTS AND REFLECT REQ CERTAIN VEH +<br>NO U-TURN SIGNS—OBEDIENCE TO +<br>REFLECTORS—COLORS AND DISTANCE SEEN +<br>SIGNAL LT—TOWED TRAILER/TOWING VEHICLE +<br>SIGNALS—HAND AND ARM—METHOD +<br>SIGNALS OF INTENTION—BY LAMP/HAND/ARM +<br>STOPPING SUDDENLY WITHOUT SIGNAL +<br>TINTING WINDOWS—RESTRICTED VIEW +<br>TOW CAR EQUIP—FLARES/WARNNG LTS/SIGNS +<br>TURN SIGNAL REQ—100′ CITY/300′ FREEWAY +<br>TURN SIGNALS +<br>UNSAFE TURN WITHOUT APPROPRIATE SIGNAL +<br>LANE CHANGE—MARKED HWY—AFTER SIGNAL +<br>LANE DIRECTIONAL CONTROL SIGNAL +<br>OBEDIENCE—NO LEFT/RIGHT TURN SIGNS +<br>OBEDIENCE—RAILWAY SIGNALS/SIGNS +<br>LICENSE PLATE LIGHT +<br>TURN SIGNAL REQ—100\′ CITY/300\′ FREEWAY |
| 5 | Seat and Belt | CARRY PASSENGERS—SEAT AND FOOTRESTS REQ +<br>DRIVER MST USE SEAT BLTS AND SHOLDR HARN +<br>ILLEGAL RIDING—NOT ON PASSENGER'S AREA +<br>MORE THAN 3 PERSONS IN FRONT SEAT<br>MTRCYCL—NO THONGS/SANDALS/OPEN TOES +<br>PASSENGER MUST USE SEAT BELT +<br>VEH MUST HAV SEAT BLTS AND SHOLDR HARNES + |
| 6 | Minor/Underage | DRIVER <21 YEARS OLD—DRIVING INDUS VEH +<br>PERMITTING UNLICENSED MINOR TO DRIVE +<br>PERMITTING UNLICENSED PERSON TO DRIVE +<br>LEAVING CHILD UNATTENDED IN MOTOR VEHICLE +<br>DEVICE TO RESTRAIN CHILD UNDR 5YR/40LB +<br>DEVICE TO RESTRAIN CHILD UNDR 6YR/60LB + |
| 7 | Lane rule | DISREGARD ONE-WAY ST/ROTARY TRAF ISLAND +<br>DIVIDED HWY—DRIVE ON RIGHT OF ROADWAY +<br>DRIVE MORE THAN 200′ IN 2-WAY TURN LN +<br>DRIVING ON RIGHT HALF OF ROADWAY +<br>DRIVING ON SIDEWALK +<br>DRIVING THROUGH SAFETY ZONES +<br>FAILURE TO DRIVE IN TRAVEL LANE +<br>HOV—CAR POOL LANES +<br>LEFT TURN FROM ONE WAY ROADWAY +<br>LEFT TURN TO ONE WAY ROADWAY +<br>LEFT TURN—POSITION/METHOD AT INTERSECT +<br>MTRCYCL NOT TO BE DRIVEN ON SIDEWALK +<br>ON DIVIDED HIGHWAY—TURN ACROSS MEDIAN +<br>ONE WAY STREET +<br>REMOVE BARRIER—DRIVE ON CLOSED HIGHWAY +<br>RIGHT OF WAY FROM PRIVATE DRIVE/ROAD +<br>RIGHT OF WAY FROM YIELD SIGN +<br>RIGHT OF WAY—ALLEY/DRIVEWAY/BUILDING +<br>RIGHT OF WAY—PASS VEH STOPPED FOR PED +<br>RIGHT OF WAY—PEDESTRIANS IN CROSSWALK +<br>RIGHT OF WAY—RIGHT TURN ON RED SIGNAL + |

**Table A1.** *Cont.*

| New Value | New Label | Old Label (of Charge) |
|---|---|---|
| 7 | Lane rule | RIGHT OF WAY—UNCONTROLLED INTERSECTION + <br> RIGHT TURN—POSITION/METHOD AT INTERSEC + <br> RIGHTS OF WAY FROM STOP SIGNS + <br> STOP IN TRAFFIC LANE + <br> STREETS OTHER THAN TRUCK ROUTES + <br> TWO WAY TURN LANE—FOR LEFT TURNS ONLY + <br> U-TURN AT INTERSEC WITH TRAF CONT DEV + <br> U-TURN IN FRONT FIRE STATION DRIVEWAY + <br> U-TURNS—BUSINESS DISTRICT OR UNSAFE + <br> UNSAFE TURNING MOVEMENT—LANE CHANGE + <br> VEHICLE TURNING LEFT AT INTERSECTION + <br> MTRCYCL RIGHT TO FULL USE TRAFFIC LANE <br> RGT OF WAY INTRSECTION—STOP <br> DRIVE MORE THAN 200\' IN 2-WAY TURN LN |
| 8 | Pass/Overtake | NO PASSING 100' INTERSECTION/CURVE/ETC + <br> NO PASSING ZONES—YELLOW LINE/MARKINGS + <br> PASS VEH IN OPPOSITE DIR —KEEP RIGHT + <br> PASS/OVERTAKE VEH ON LEFT—METHOD + <br> PASS/OVERTAKE VEH ON RIGHT—METHOD + <br> PASSING BETWEEN STOPPED OR MOVING VEH + <br> UNLAWFULLY OVERTAKE AND PASS VEHICLE IN <br> SCHOOL ZONE + |
| 9 | Reckless driving general | BLOCK INTERSECTION—OBSTRUCT PASSAGE + <br> DRIVRS VIEW OBST/PASSENGER INTERFER + <br> FOLLOWING TOO CLOSE + <br> IMPROPER OR PROHIBITED U-TURN IN SCHOOL ZONE + <br> RECKLESS DRIVING + <br> WILLFUL OBSTRUCTION/DELAY OF TRAIN <br> WRITTEN REPORT BY DRIVERS/OWNERS <br> W/INJURY/PROP DAM |
| 10 | Careless driving general | BOARD OR ALIGHT FROM MOVING VEHICLE + <br> DRIVING IN A CARELESS MANNER + <br> DRIVERS POSITION WHILE OPERATING <br> FAIL USE DUE CARE AVOID COLL W/PED + <br> FULL ATTENTION TO DRIVING + <br> HAND POSITION OF DRIVER + <br> HORNS—UNNECESSARY USE + <br> NO TV RECEIVER VISIBLE AT DRIVERS SEAT + <br> OPEN DOOR IN TRAFFIC—LEAVE DOOR OPEN + <br> UNATTENDED VEH—REMOVE KEY/STOP ENGINE + <br> REMOVE KEY/LOCK IGNITION/LEAVE ENG RUN + <br> UNSAFE BACKING + <br> UNSAFE LOAD + <br> UNSAFE STARTING A STOPPED VEHICLE + |
| 11a | Yield to Car | FAIL TO YEILD ON FLASHING YELLOW ARROW + <br> FAIL TO YIELD ON FLASHING YELLOW ARROW + <br> FAIL TO YIELD TO ONCOMING TRAFFIC + <br> FAIL TO YIELD TO PERSON RIDING BICYCLE + <br> YIELD SIGN—DISREGARD OF + |
| 12/11b | Yield to Pedestrian | FAILURE TO YIELD TO A PEDESTRIAN |
| 13/11c | Yield to emergency Vehicle | CROSSING FIRE HOSE + <br> FOLLOW FIRE TRUCKS—PARK W/IN 300'–500' + <br> YIELD TO EMERGENCY VEHICLE + |

**Table A1.** *Cont.*

| New Value | New Label | Old Label (of Charge) |
|---|---|---|
| 14 | Red Light Stop | DRIVE THRU PRIV PROP TO AVOID RED LITE + <br> RED FLAGS/LIGHTS—EXTENDED LOAD/LOAD>4′ + <br> RED LIGHTS AND SIRENS MAY REMOVE/DESTROY + <br> RED TRAFFIC SIGNAL—POSITION/METHOD + <br> STOP LIGHTS + <br> STOP REQ SIGNAL POL OFF—ATTEMPT ELUDE + <br> STOP SIGN—POSITION/METHOD OF STOP + |
| 15 | Plate and car registration | EXPIRED LICENSE PLATES + <br> FAIL TO SURRENDER SUSPENDED LICENSE PLATES + <br> FICTITIOUS/ALTERED—PLATES/CERTIFICATE + <br> FICTITIOUS/SUSP/REV VEH REGISTRATION + <br> LICENSE PLATES—METHOD OF DISPLAY + <br> NO NV LIC PLTS W/IN 30 DAYS RESIDENCY + <br> NO NV LIC PLTS W/IN 60 DAYS RESIDENCY + <br> NO REGISTRATION IN VEHICLE + <br> NO VEHICLE REGISTRATION + <br> OPERATE UNTREGISTERED MOPED + <br> OPERATE VEH W/ALT VEH NUM—SERIAL NUM + <br> PERMIT TO OPERATE UNREGISTERED VEHICLE + <br> REVOKED REGISTRATION + <br> SUSPENDED REGISTRATION/PLATES + <br> FAIL SUR PLATES/DL TO DEPT OF MTR VEH + <br> UNREGISTERED VEHICLE/TRAILER/SEMI TRAILER + <br> ADDRESS CHANGE—REG—WITHIN 10 DAYS + <br> ADDRESS CHANGE—REG—WITHIN 30 DAYS + <br> UNLAWFUL TO LEND LICENSE PLATES OR REG |
| 16 | Speed Limit | BASIC RLE—FSTER/POSTED 1–10 OVER + <br> BASIC RLE—FSTR/POSTED 11–15 OVER + <br> BASIC RLE—FSTR/POSTED 16–20 OVER + <br> BASIC RLE—FSTR/POSTED 21 OR OVER + <br> BASIC RLE—FSTR/POSTED 21–30 OVER + <br> BASIC RLE—FSTR/POSTED 31–40 OVER + <br><br> BASIC RLE—FSTR/POSTED 41+ OVER + |
| 17 | Speed on Conditions | BASIC SPEED—TOO FAST FOR CONDITIONS + <br> FAIL DECREASE SPD AND USE DUE CARE + <br> MANR/RAT/SPD—ENDGR LIF/LIMB/PROP—CRLES + <br> MINIMUM SPEED—IMPEDE OR BLOCK TRAFFIC + <br> MISDEMEANOR + <br> SPEED CONTEST + <br> IMPEDE TRAFFIC/TOO SLOW—MOVE TO RIGHT <br> PRIMA FACE SPD—SCH CROSS 1–15 OVER + <br> PRI/FACIE SPD—SCH ZN 16MPH/LIMIT <br> PRIMA FACE SPD—SCH ZONES—15 MPH <br> SLOW TRAFFIC TO DRIVE IN RIGHT LANE + |
| 18 | Parking | VIOLATION PARK RULES ALCOHOL + <br> HANDICAPPED PARK ONLY (PERMIT) + <br> PARK TO SELL/WASH/GREASE/REPAIR VEH <br> PARKING ADJACENT TO SCHOOL WHEN POSTED <br> PARKING IN A TAXI OR BUS ZONE <br> PARKING IN RED ZONE + <br> PARKING NEAR FIRE HYDRANT + <br> PROHIB PARK GEN—SIDEWALK/DRIVEWAY/ETC + <br> STOP/STAND/PARK IN BUS STOP/TAXI STAND ZONE + <br> VIOLATION PARK RULES ALCOHOL + |

**Table A1.** *Cont.*

| New Value | New Label | Old Label (of Charge) |
|---|---|---|
| 18 | Parking | VIOLATE PARK RULES + <br> PARK IN ALLEY, INCLUDING BLOCKING DRIVEWAY + <br> PARKING—OVER 18" FROM CURB FACE <br> PROJECT INTO ST—NOT > 15 IN FROM CURB <br> SET BRAKE—TURN WHEELS TO CURB ON GRADE <br> STOP/STAND/PARK IN HAZARD/CONGEST AREA |
| 19 | Duty | DUTY APPROACHING EMERGENCY VEHICLE + <br> DUTY GIVE INFORMATION AND RENDER AID + <br> DUTY TO STOP—PROPERTY DAMAGE ONLY <br> DUTY UPON ACCIDENT W/INJURY OR DAMAGE + <br> DUTY UPON DAMAGING UNATTENDED VEH/PROP + <br> IMMEDIATE REPORT OF ACCIDENT TO POLICE + <br> STOP FOR SCHOOL CROSSING GUARDS + |
| 20 | Power Showing | AGGRESSIVE DRIVING + <br> EXHAUST SYSTEM—MODIFIED TO MAKE NOISE + <br> EXHIBITION OF POWER + <br> OBEDIENCE AUTHORIZED FLAGMAN—SIGNAL + <br> OBEDIENCE TO TRAFFIC CONTROL DEVICE + <br> OBSTRUCTING TRAFFIC + |
| 21 | School bus | STOPPING FOR SCHOOL BUS |
| 22 | Tax/Truck and other Compliance | BEHIND ON LOG BOOK <br><br> FAILURE TO COMPLY W/TAXI STAND USE + <br> PROHIBITED/REQUIRED ACTS—TAXI DRIVER <br> TRIP SHEET—CAB + <br> TAXICAB STANDS—SEEKING FARE OR PARKING <br> HEIGHT—VEHICLE DISTANCE FROM ROADWAY <br> EQUIPMENT VIOLATION + <br> ABANDONED VEHICLE |
| 23 | DUI and Alcohol | DRINK INTOXICATING LIQUOR WHEN DRIVING <br> DUI DRUGS CHEMICALS ORGANIC SOLVENT <br> DUI LIQUOR <br> DUI LIQUOR AND/OR DRUGS <br> DUI SUBSEQUENT ARREST <br> MINOR IN CONSUMPTION OF ALCOHOL <br> MINOR IN POSSESSION OF ALCOHOL + <br> OPEN ALCOHOLIC CONTAINER IN VEHICLE <br> POSS/CNSM ALCOHOL ON PED MALL FROM <br> GLASS/METAL/ORG + |
| 24 | Non-Traffic Charges | ACTIONS WHICH CONSTITUTE THEFT + <br> ADEQUATE WATER FOR ANIMALS + <br> AFFRAY + <br> AID AND ABET A PROSTITUTE + <br> ANIMAL VACCINATION CERTIFICATE REQ + <br> ASSAULT + <br> ATMPT SMOKE/CONSUME MARI IN PUBLIC <br> PLACE/VEH/STORE + <br> BATTERY DOMESTIC VIOLENCE, FIRST OFFENSE + <br> BATTERY + <br> BATTERY DOMESTIC VIOLENCE—2nd Offense + <br> BATTERY DOMESTIC VIOLENCE—2ND OFFENSE + <br> BATTER DOMESTIC VIOLENCE, FIRST OFFENSE <br> BATTER DOMESTIC VIOLENCE, SECOND OFFENSE <br> BATTERY DOMESTIC VIOLENCE, SECOND OFFENSE + <br> BATTERY/DOMESTIC VIOLENCE + |

Table A1. *Cont.*

| New Value | New Label | Old Label (of Charge) |
|---|---|---|
| 24 | Non-Traffic Charges | BUSINESS LICENSE VIOLATION + <br> CARRY CONCEALED WEAPON W/O PERMIT + <br> COERCION + <br> COMMIT ACT/INTERFERE W/PEACEFUL CONDCT + <br> CONSUME ALCHOL ON PREMISE OFF/SALE ONLY + <br> CONTRIBUTING TO DELINQUENCY OF MINOR + <br> CONVICTED PERSON FAIL TO REGISTER + <br> CONVICTED PERSON FAIL/CHANGE ADDRESS + <br> CREATE DISTURBANCE IN SCHOOL + <br> CRUELTY TO ANIMALS + <br> DEFECATING IN PUBLIC + <br> DEFRAUDING AN INNKEEPER+ <br> DEFRAUD CAB DRIVER+ <br> DESTRUCTION PRIVATE PROPERTY <br> DISCHARGE OF A FIREARM + <br> DISTURBING THE PEACE + <br> DO BUSINESS WITHOUT A LICENSE + <br> DOG RUNNING AT LARGE + <br> DRAW A DEADLY WEAPON + <br> EMBEZZLEMENT + <br> EMISSIONS OF SMOKE, STEAM OR FUMES + <br> FAIL TO FURNISH INFO TO ANIMAL REGULATORY OFFICER + <br> FAIL TO REGISTER GARAGE + <br> FAILURE TO LICENSE DOG/CAT + <br> FAILURE TO RESTRICT ANIMAL <br> FALSE REPORT OF A CRIME + <br> FTA—ORDER TO SHOW CAUSE/WITNESS + <br> GARBAGE REMOVAL + <br> GIVE FALSE INFO TO PUBLIC OFFICER, GIVE OR LEAVE MARIJUANA TO PERSON UNDER 21 + <br> GRAFFITI + <br> GRAFFITI IMPLEMENTS WITH INTENT TO VAND + <br> HANDLEBAR HEIGHTS + <br> HARASSMENT + <br> INHALE GLUE/OTHER CHEMICAL + <br> INTERFERENCE WITH GARBAGE CONTAINER <br> JUNKAGE/DEAD STORAGE—MORE THAN 24 HRS <br> LEWD AND LASCIVIOUS BEHAVIOR + <br> LEWD EXPOSURE + <br> LITTERING + <br> LODGING IN A PASSENGER CAR <br> LODGING WITHOUT CONSENT + <br> LOITERING ABOUT A SCHOOL + <br> LOITERING ABOUT SCHOOL/PLACE CHILDREN CONGREGATE + <br> LOITERING FOR PURPOSE OF PROSTITUTION + <br> MAINTAINING A PUBLIC NUISANCE + <br> MALICIOUS DESTRUCTION OF PROPERTY + <br> MALICIOUS PROSECUTION + <br> MINOR GAMBLING <br> MINOR IN CASINO <br> MISUSE OF BUS SHELTER BENCH <br> NOISE DISTURBANCE + <br> OBEY ORDER/DIRECTION OF PUBLIC OFFICER + <br> OBSTRUCTING/FALSE INFO TO P. O. + <br> RESISTING PUBLIC OFFICER <br> OPEN CONTAINER AT BUS SHELTER <br> OBTAIN MONEY UNDER FALSE PRETENSES + |

Table A1. *Cont.*

| New Value | New Label | Old Label (of Charge) |
|---|---|---|
| 24 | Non-Traffic Charges | PERSONAL MARIJUANA CULTIVATION LAWS 1ST OFFENSE + GLASS/METAL/ORG + PARK BICYCLES—BLOCK PEDESTRIAN TRAFFIC POSSESS FIREARM U/INFLUENCE DRUGS/ALC + POSSESS HYPODERMIC DEVICE + POSSESS LESS THAN 1 OUNCE OF MARIJUANA, POSSESS OPEN LIQUOR ON PLAYGROUND, POSSESS UNREGISTERED FIREARM, POSSESSION FIREWORKS, POSSESSION OF A CONTROL SUB IMITATION, POSSESSION OF A SHOPPING CART + POSSESSION OF STOLEN PROPERTY + PROHIBITED CONTAINER + PROVOKE OR ATTEMPT TO PROVOKE BREACH OF PEACE + PROVOKING BREACH OF PEACE + REMAIN IN PARK AFTER HOURS OF CLOSURE + ROLLER SKATES, ETC—ILLEGAL ON ROADWAY + SALE/FURNISH LIQUOR TO MINOR + SMOKE/CONSUME MARIJUANA IN PUBLIC PLACE/VEH/STORE + SMOKING UNLAWFUL IN PUBLIC PLACES + SOLICITING WITHOUT A PERMIT + SPAY AND NEUTER + SPITTING ON SIDEWALK + STALKING + SPILL LOAD ON HWY OR ST/COVERED LOAD + STANDARDS OF CONDUCT WHILE ON DUTY + RIDE BICYCLE—1 PERSON UNLESS EQUIPPED + RESTRICTED USE BY BICYCLE/PED./MOPED + STREET PREFORMER—DESIGNATION LOCATION + PETIT LARCENY + TAMPERING/INJURING A VEHICLE + THREATEN TO HARM STUDENT OR SCHOOL EMPLOYEE, THREATENING PHONE CALL + THROW BURNING OBJECT + UNLAWFUL PRESENCE IN A CHILDRENS PARK + UNLAWFUL PROSTITUTION RELATED ACTIVITY, UNLAWFUL TRANS/USE—DEALER REGISTRATI, UNLAWFUL USE OF CELL PHONE OR HANDHELD DEVICE, UNLAWFUL USE/POSSESSION OF DRUG PARAPHERNALIA, UNIFORM FIRE CODE , UNLAWFUL ACTS—STREET PERFORMERS + UNLAWFUL COMMUNICATION/EXCHANGE WITH PRISONER + URINATING IN PUBLIC + VEHICLE CONFINEMENT—ANIMALS + VIOL DOM VIOLENCE TPO + VIOLATE RESTRAINING ORDER, WALK IN ROADWAY WHEN SIDEWALK PROVIDED, WEAPON IN PARK, NON-RESIDENT DRIVE ON CANCELLED DL, NON-RESIDENT DRIVE ON REVOKED LIC, NON-RESIDENT DRIVE ON SUSP/CANC/REV DL, NON-RESIDENT DRIVE ON SUSPENDED LIC, BICYCL—LIGHTS/REFLECTRS/BRKES/WARN DEV BICYCL—PARENT/GUARDN—ENSURE RULES OBEY BICYCL—RIDE ON RIGHT SIDE OF ROADWAY BICYCL—SPEED TOO FAST FOR CONDITIONS BICYCL—TRAFFIC SIGNALS TO BE OBEYED |

**Table A1.** *Cont.*

| New Value | New Label | Old Label (of Charge) |
|:---:|:---:|:---|
| 24 | Non-Traffic Charges | BICYCLE—CARRYING ARTICLES<br>BIKE/SKATEBOARD ON SIDEWLK IN BUS DIST<br>BICYCL—ENTER/EMERGE—ALLEY/DRIVWY/BLDG<br>HDGEAR—GLASSES/SHIELDS HELMET FASTENED<br>PED CROSSING NOT IN CROSSWLK—JAYWALK<br>PEDESTRIAN FAIL TO USE CROSSWALK<br>PEDESTRIAN MUST USE RIGHT OF CROSSWALK<br>PEDESTRIAN OBSTRUCTING SIDEWALK<br>PEDESTRIAN ON HIGHWAY WHERE PROHIBITED<br>PEDESTRIAN OR VEHICULAR INTERFERENCE<br>PEDESTRIAN SHALL YIELD<br>PEDESTRIAN SOLICITING ON HIGHWAY<br>PEDESTRIAN UNDER INFLUENCE ON ROADWAY<br>PEDESTRIANS TO OBEY TRAFFIC SIGNALS +<br>PEEPING +<br>PROHIBIT CAMPING/LODGING/ETC W/PUBLIC<br>RIGHT OF WAY<br>RESTRICTED ACCESS<br>TRESPASSING, TRESSPASS/LOITER/COMMIT<br>NUISANCE ON/NEAR SCHOOL<br>HITCHHIKE/SOLICIT BUSINESS FROM DRIVER +<br>OBTAIN PROPERTY UNDER FALSE PRETENSES<br>SKATES, ETC—ILLEGAL FREMONT SIDEWALK<br>SOLICITING CERTAIN LOCATIONS<br>THROWING DEADLY MISSILES<br>UNLAWFUL ACTS/ANIMALS<br>UNLAWFUL USE FALSE IDENTIFICATION |

**Table A2.** 24 new charge categories grouped by charge type into 8 categories.

| | 8 New Value Labels | 24 Old Value Labels |
|:---:|:---:|:---|
| 1 | Insurance | Insurance |
| 2 | Driver license | Driver license |
| 3 | Vehicle Conditions | Vehicle conditions<br>Lamps, Light Color/Signals |
| 4 | Moving Violations | Seat and Belt<br>Minor/Underage<br>Lane rule<br>Pass/Overtake<br>Reckless and Careless driving<br>Yield to car or pedestrian or emergency or school bus<br>Red Light Stop<br>Speed Limit<br>Duty<br>Power Showing<br>Tax/Truck and other Compliance |
| 5 | Vehicle Registration | Plate and car registration |
| 6 | Parking | Parking |
| 7 | DUI | DUI and Alcohol |
| 8 | Non-Traffic | Non-Traffic |

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
