# Peer review of "Costs and Consequences of Traffic Fines and Fees: A Case Study of Open Warrants in Las Vegas, Nevada"

_socsci, doi:10.3390/socsci10110440_

Round 1
Reviewer 1 Report
By examining traffic violations and related warrants, this manuscript explores the disproportionate impact of traffic fines and fees. In particular, the manuscript uses Las Vegas Municipal Court data to illustrate how fees and fines related to traffic violations disproportionately impact certain demographic groups. While this is a well-written paper exploring a unique and important topic, there are some questions and concerns outlined below that should be considered. Hopefully the author(s) find the comments useful.
Major:
The research questions for this study were not clear, even after reading the results. On page 1, the author mentions that “there is little research on race and income disparities in traffic stops…” while also indicating the study is about “examin[ing] the impact of reliance on traffic fines and fees…” Later, in the Methods section, the author mentions that the study aims to explore how the problem of traffic fines and fees has changed over time. More clearly defining the purpose of the research will help contextualize the analyses related to, for example, age, gender, and categories of charges.
The theoretical or practical implications were not immediately clear. For example, the findings do not speak to why costly traffic fines and fees are a prominent feature of the criminal-legal system, a debate which is set up in the literature review. Additionally, although the manuscript provides an excellent description of mass incarceration as a racialized system of control, it is not clear how the findings contribute to that idea; the findings related to race only account for one of the six main findings. Finally, given Assembly Bill AB116, it is unclear what these findings mean for post-2023.
The discussion of the legislative history of monetary sanctions in Nevada seems more appropriate for the introduction. Although the author mentions that reports and testimony were examined, there is no systematic analysis of those documents. Rather, the discussion of the legislative timeline seems to set up the problem underlying the study.
It would be helpful to have a more detailed explanation of the Methods. For example, why use warrants to assess disparities in fines and fees? Is it possible that these analyses are excluding, for example, older, White women who are paying fines and, therefore, do not have a warrant issued? Also, why was the 2011-2021 timeline used? Greater detail for the Data Analysis section would also be helpful.
I was surprised by the some of the descriptive analyses, including gender and age distributions as well as the breakdown between administrative and behavioral citations. Given the emphasis on a racialized system of mass incarceration, I expected more analyses related to that point. For example, the author could have examined the probability of a fine (or warrant), given a person’s racial background. The author could have also examined the interaction between race and the other demographic characteristics to highlight ways that issues involving race can be exacerbated.
The author states that even though the data set is incomplete there is “no evidence that it would be dramatically different” (p. 15, lines 435-436) if a more complete data set was used. Please explain why this is the case.
Minor:
It was unclear if sections 3.1.2 and 3.13 were referring to different sets of data.
There is a missing “to” in line 341.
There is an extra space between words on line 513.
Author Response
In response to Reviewer 1:
Thank you so much for your comments!
R1.1 Comment: The research questions for this study were not clear, even after reading the results. On page 1, the author mentions that “there is little research on race and income disparities in traffic stops…” while also indicating the study is about “examin[ing] the impact of reliance on traffic fines and fees…” Later, in the Methods section, the author mentions that the study aims to explore how the problem of traffic fines and fees has changed over time. More clearly defining the purpose of the research will help contextualize the analyses related to, for example, age, gender, and categories of charges.
Response: Thank you for your comments. We have made our research questions more explicit in the abstract and introduction, more clear in findings subheadings, and more consistent throughout the document.
The statement referred to on page 1 has been changed to say that highlights our emphasis on demographic differences in warrants issued: “While research is beginning to examine racial disparities in traffic stops, and in monetary sanctions, there is little research on demographic differences, especially economic inequality, in the traffic warrants generated by failure to pay.”
Unfortunately we could not find where we said the study will explore how the problem has changed over time. We hope that the other changes took care of this problem and if we missed something please let us know.
R1.2 Comment: The theoretical or practical implications were not immediately clear. ​​For example, the findings do not speak to why costly traffic fines and fees are a prominent feature of the criminal-legal system, a debate which is set up in the literature review.
Response: The literature review provides the various theories for why costly traffic fines and fees may be prominent in funding the legal system. Our finding that the Nevada legislature explicitly increased fines and fees (RQ2) to raise revenues and fund services in a context of dramatic population growth speaks to why this has become a prominent feature in Nevada. We have made this finding more prominent and directly addressed this in the discussion and conclusion.
We believe that making this more explicit as a research question helps explain the context and thus better addresses the practical and theoretical implications for our study.
R1.3 Comment: Additionally, although the manuscript provides an excellent description of mass incarceration as a racialized system of control, it is not clear how the findings contribute to that idea; the findings related to race only account for one of the six main findings.
Response: We reorganized the subheadings and discussion of our findings in the hopes of highlighting the importance of our two main findings, economic and racial inequality.
R 1.4 Comment: Finally, given Assembly Bill AB116, it is unclear what these findings mean for post-2023
Response: We edited our discussion to cover the fact that AB116 may slow arrests, but the problems in taxation by citation still exist.
R1.5 Comment: The discussion of the legislative history of monetary sanctions in Nevada seems more appropriate for the introduction. Although the author mentions that reports and testimony were examined, there is no systematic analysis of those documents. Rather, the discussion of the legislative timeline seems to set up the problem underlying the study.
Response: We agree to a certain extent, and debated the appropriate place for the analysis of the legislative history. Ultimately, we realized that making this more prominent as a research question helped address the concern about theoretical and practical implications, better tied to our literature review, and answered the concern in the second comment about why costly fines and fees are such a prominent feature.
R1.6 Comment: It would be helpful to have a more detailed explanation of the Methods. For example, why use warrants to assess disparities in fines and fees? Is it possible that these analyses are excluding, for example, older, White women who are paying fines and, therefore, do not have a warrant issued?
Response: Our initial draft was unclear that in Nevada, failure to pay traffic violations turns into a bench warrant. For that reason, we wanted to find out who is being issued these warrants. We have made that explicit in the beginning of the methods section.
Data on who is assessed fees is not publicly available. We discuss the need to have data on differences between who are issued tickets and who can’t pay in the future research section.
R1.7 Comment: Also, why was the 2011-2021 timeline used?
Response: The court only made data from 2011-2021 available for extraction from their website. Anything before 2011 was unavailable. Our analysis focused on warrants that the court issued for the period from 2012 to 2020. We excluded warrant data for the years 2011 and 2021 as it was partial and we observed that the warrants were not representative enough at the time of data extraction in January 2021. We have made this clear in the methods section.
R1.8 Comment: Greater detail for the Data Analysis section would also be helpful.
Response: We are hoping that the changes we have made in response to other comments have made this section more clear.
R1.9 Comment I was surprised by the some of the descriptive analyses, including gender and age distributions as well as the breakdown between administrative and behavioral citations. Given the emphasis on a racialized system of mass incarceration, I expected more analyses related to that point. For example, the author could have examined the probability of a fine (or warrant), given a person’s racial background. The author could have also examined the interaction between race and the other demographic characteristics to highlight ways that issues involving race can be exacerbated.
Response: It would be great to examine the interaction of race with other demographic characteristics like gender, age, etc. However, this is outside of the scope of this project. We hope that we or others will expand this project by looking at the interaction of demographic factors in future.
R1.11 Comment: The author states that even though the data set is incomplete there is “no evidence that it would be dramatically different” (p. 15, lines 435-436) if a more complete data set was used. Please explain why this is the case.
Response: We have deleted that phrase and clarified our explanation for why we just had one court in our data.
R1.12 Comment: It was unclear if sections 3.1.2 and 3.13 were referring to different sets of data.
Response: They are referring to the same dataset. These sections have been reorganized for clarity. Parts of section 3.1.2 was moved to the introduction which also makes the practical implications of fines and fees more immediately clear.
We cleared up the editorial comments.
Reviewer 2 Report
The reviewed paper deals with the phenomenon of “taxation by citation” in Nevada. It must be emphasized that the very phenomenon of exaggerated imposition of traffic tickets and fines is not limited to the United States, but is also noticeable in Europe, especially in the countries of Central and Eastern Europe (including Poland). Therefore, the reviewed paper is important not only for the American reader. This article offers a very thorough analysis of the phenomenon in relation to monetary sanctions in Nevada. Moreover, it is well-written in terms of language and style. The authors’ original analyzes were preceded by citing the results of previous studies (pp. 2–4). The literature used here should be considered sufficient.
The Authors presents the finding that: “The current system of ‘taxation by citation’ in Nevada is extremely inequitable, disproportionately impacting Black citizens in particular” (p. 15). Furthermore: “… the current fines and fees syten in Nevada, as in other states, is consequence of the federal government shifting the cost burden of local services to states who are otherwise uboprepared to raise funds…” (p. 17). This view is adequately justified in the paper. It must be emphasized that the use of fines and fees system as the source of funds for local authorities is not a US speciality.
As for the critical remarks, the authors should justify more broadly why they used data from only one court, the Las Vegas Municipal Court (p. 4).
Author Response
In response to Reviewer 2:
Thank you so much for your suggestions and for your kind comments regarding our analysis.
R2.1 Comment: The reviewed paper deals with the phenomenon of “taxation by citation” in Nevada. It must be emphasized that the very phenomenon of exaggerated imposition of traffic tickets and fines is not limited to the United States, but is also noticeable in Europe, especially in the countries of Central and Eastern Europe (including Poland).
AND It must be emphasized that the use of fines and fees system as the source of funds for local authorities is not a US speciality.
Response: This phenomenon has been recognized as global in the introduction and conclusion.
R2.2 As for the critical remarks, the authors should justify more broadly why they used data from only one court, the Las Vegas Municipal Court (p. 4).
Response: We discuss this in the limitations section 6.1 and have added more to the methods discussion. The Las Vegas Municipal Court was the only court system serving the largest metropolitan area in Nevada, Las Vegas, that could provide data. The other local court systems, Las Vegas Justice Court and those in North Las Vegas, have antiquated case management systems and are unable to pull their data into a report for analysis. We hope that as datasets become available, we can duplicate these efforts and compare analyses.
Reviewer 3 Report
Lack of hypothesis.
I propose to give the definition of marginalized
communities and explain the difference between poor and marginalized
communities.
Author Response
In response to Reviewer 3:
Thank you so much for your comments.
R3.1 Comment: Lack of hypothesis
Response: A statement of our research questions is now made clear immediately in the introduction and in the methods sections.
R3.2 Comment: I propose to give the definition of marginalized communities and explain the difference between poor and marginalized
Communities
Response: Throughout the document, we specified the communities rather than just describing them as marginalized. Our focus is not more clearly on economic inequality and poor communities, and Blacks and communities that suffered racial discrimination.
Round 2
Reviewer 1 Report
Although I still have some concerns with the analysis and discussion of the legislative history because it seems more like context and not data analysis, the findings related to fines and fees in Las Vegas are much more clear. The re-formatting and revising of the manuscript also helps distinguish the research questions and purpose of the research.